# Keratinocyte PIEZO1 modulates cutaneous mechanosensation

Alexander R Mikesell, Olena Isaeva, Francie Moehring, Katelyn E Sadler, Anthony D Menzel, Cheryl L Stucky*

Department of Cell Biology, Neurobiology and Anatomy, Medical College of Wisconsin, Wauwatosa, United States

**Abstract** Epidermal keratinocytes mediate touch sensation by detecting and encoding tactile information to sensory neurons. However, the specific mechanotransducers that enable keratinocytes to respond to mechanical stimulation are unknown. Here, we found that the mechanically-gated ion channel PIEZO1 is a key keratinocyte mechanotransducer. Keratinocyte expression of PIEZO1 is critical for normal sensory afferent firing and behavioral responses to mechanical stimuli in mice.

## Editor's evaluation

Although sensory neurons are thought to be the primary detectors of environmental stimuli in skin, it is more and more appreciated that non-neuronal cell types also play important roles. This study investigates whether a very common type of cell in the skin functions in touch sensation and identifies the mechanically gated ion channel Piezo1 as key gene.

*For correspondence:
cstucky@mcw.edu

**Competing interest:** The authors declare that no competing interests exist.

## Introduction

Despite the importance of touch sensation for daily life, we are only beginning to understand the molecular and cellular signaling mechanisms through which tactile information is transduced from the skin to the central nervous system. In the last decade, sensory biologists have determined that non-neuronal cells and specialized end organ structures in the skin interact with sensory neurons to mediate touch sensation; Merkel cells and Meissner corpuscles encode unique aspects of gentle touch by tuning the responses of Aβ sensory neurons, and specialized terminal Schwann cells modulate the firing of nociceptors to noxious touch (*Maksimovic et al., 2014*; *Woo et al., 2014*; *Hoffman et al., 2018*; *Abdo et al., 2019*; *Ojeda-Alonso et al., 2022*; *Neubarth et al., 2020*). Keratinocytes, which constitute >95% of the cells in the epidermis, are innately sensitive to mechanical force, are capable of releasing a wide array of neuroactive factors, and form close 'synapse-like' connections with intraepidermal nerve fibers (*Fuchs, 1995*; *Koizumi et al., 2004*; *Tsutsumi et al., 2009*; *Goto et al., 2010*; *Lumpkin and Caterina, 2007*; *Hou et al., 2011*; *Shi et al., 2013*; *Barr et al., 2013*; *Talagas et al., 2020a*; *Talagas et al., 2020c*). Moreover, optogenetic activation of keratinocytes induces action potential firing in sensory neurons, whereas optogenetic inhibition of keratinocyte activity decreases both sensory neuron and behavioral responses to tactile stimuli (*Baumbauer et al., 2015*; *Moehring et al., 2018a*). Thus, keratinocyte activity is critical for normal sensory neuron and behavioral responses to mechanical stimuli.

The ability of keratinocytes to respond to force and contribute to touch sensation indicates that they must express one or more mechanically sensitive proteins, but the specific keratinocyte mechanotransducer(s) have not yet been identified. PIEZO1 and PIEZO2 are mechanically gated, non-selective cation channels that share approximately 42% amino acid similarity and are widely expressed

in tissues that respond to mechanical force (e.g. lung, skin, bladder, and vasculature) (*Li et al., 2014*; *Ranade et al., 2014a*; *Friedrich et al., 2019*; *Dalghi et al., 2019*). PIEZO2 expression in both dorsal root ganglia (DRG) sensory neurons and Merkel cells is required for innocuous touch sensation (*Woo et al., 2014*; *Ranade et al., 2014b*; *Coste et al., 2010*). However, it is unknown if PIEZO1 also contributes to touch sensation. Because PIEZO1 is highly expressed in mouse skin (*Coste et al., 2010*), we hypothesized that this channel may be a key mechanotransducer in keratinocytes. Here, we show that virtually all keratinocytes isolated from mouse and human skin respond to the PIEZO1 agonist Yoda1 and that PIEZO1 expression is important for keratinocyte mechanical sensitivity. Furthermore, we demonstrate that loss of epidermal PIEZO1 decreases the firing rate of sensory nerve fibers in response to mechanical stimulation of the skin and blunts behavioral responses to both innocuous and noxious mechanical stimuli in vivo. Together, these data demonstrate that epidermal PIEZO1 is critical for normal touch sensation.

## Results

To determine if PIEZO1 is a mechanotransducer in keratinocytes, we generated epidermal cell-specific PIEZO1 knockout mice (PIEZO1cKO) by crossing *Keratin14(Krt14)*[Cre] and *Piezo1*[loxp/loxp] mice (*Cahalan et al., 2015*). Successful knockout of the channel was verified with RNAscope in situ hybridization; RNA probes for *Piezo1* stained the epidermis of wild-type mice, but *Piezo1* puncta were absent from PIEZO1cKO epidermis (*Figure 1A*). Importantly, PIEZO1 deletion did not disrupt gross epidermal morphology, as the stratum corneum and stratum spinosum appeared similar in PIEZO1cKO and wild-type mice (*Figure 1—figure supplement 1A and B*). Quantitative real-time PCR also confirmed successful knockout of the channel; *Piezo1* transcript was detected in the epidermis of wild-type mice but absent in PIEZO1cKO samples (*Figure 1B*). On a functional level, in vitro calcium imaging experiments revealed that keratinocytes isolated from wild-type animals responded robustly to the PIEZO1-specific chemical agonist Yoda1 (*Syeda et al., 2015*) in a concentration-dependent manner (*Figure 1C and D*). In contrast, keratinocytes from PIEZO1cKO animals were virtually unresponsive to Yoda1 (*Figure 1C and D*). Similarly, primary human keratinocytes displayed robust, concentration-dependent intracellular calcium flux in response to Yoda1 (*Figure 1D and E*). These data indicate that functional PIEZO1 is expressed in both human and mouse keratinocytes.

### PIEZO1 mediates keratinocyte mechanical sensitivity

To verify the mechanosensitivity of keratinocytes, we performed whole cell patch clamp recordings in primary cultures of mouse keratinocytes while probing the cell membrane with increasing levels of indentation. Keratinocytes were sensitive to mechanical stimulation as the majority of cells responded to a very gentle membrane indentation (≤1 µm) with a mechanically activated (MA) current. However, the amplitude of the MA current evoked by membrane indentation did not show a clear dependence on the increase in membrane indentation depth. *Figure 2A* shows a representative example of a MA current evoked in a keratinocyte in response to stepwise increases in membrane indentation. In this recording, the initial current was induced in response to 0.50 µm membrane indentation (shown in blue), maximum current was observed in response to the next stimulation (black, 0.75 µm membrane displacement), and subsequent displacements resulted in smaller currents (red, 1.00 µm) or no current (green, 1.25 µm). Increasing the time between mechanical stimulations to 2 min did not affect the properties of keratinocyte responses to increasing mechanical indentation (data not shown). We next examined whether PIEZO1 is required for keratinocytes mechanosensitivity. PIEZO1cKO keratinocytes required greater indentation to elicit MA currents compared to wild-type cells, indicating PIEZO1cKO keratinocytes have elevated mechanical thresholds (*Figure 2B*). In addition, there was an increase in the number of keratinocytes unresponsive to membrane indentation (mechanically insensitive) in the PIEZO1cKO group (51.35%, 19 out of 37 cells) compared to wild type group (21.74%, 5 out of 23 cells) (*Figure 2C*). However, there was no change in the proportion of keratinocytes that responded to membrane indentation with rapidly adapting (RA), intermediately adapting (IA), or slowly adapting (SA) currents (*Figure 2D and E*). Additionally, we did not observe any effect of PIEZO1 deletion on the profiles of RA, IA, and SA currents. Furthermore, there was no change in the maximum current amplitude elicited by membrane indentation at any force tested (*Figure 2F*). These findings indicate that

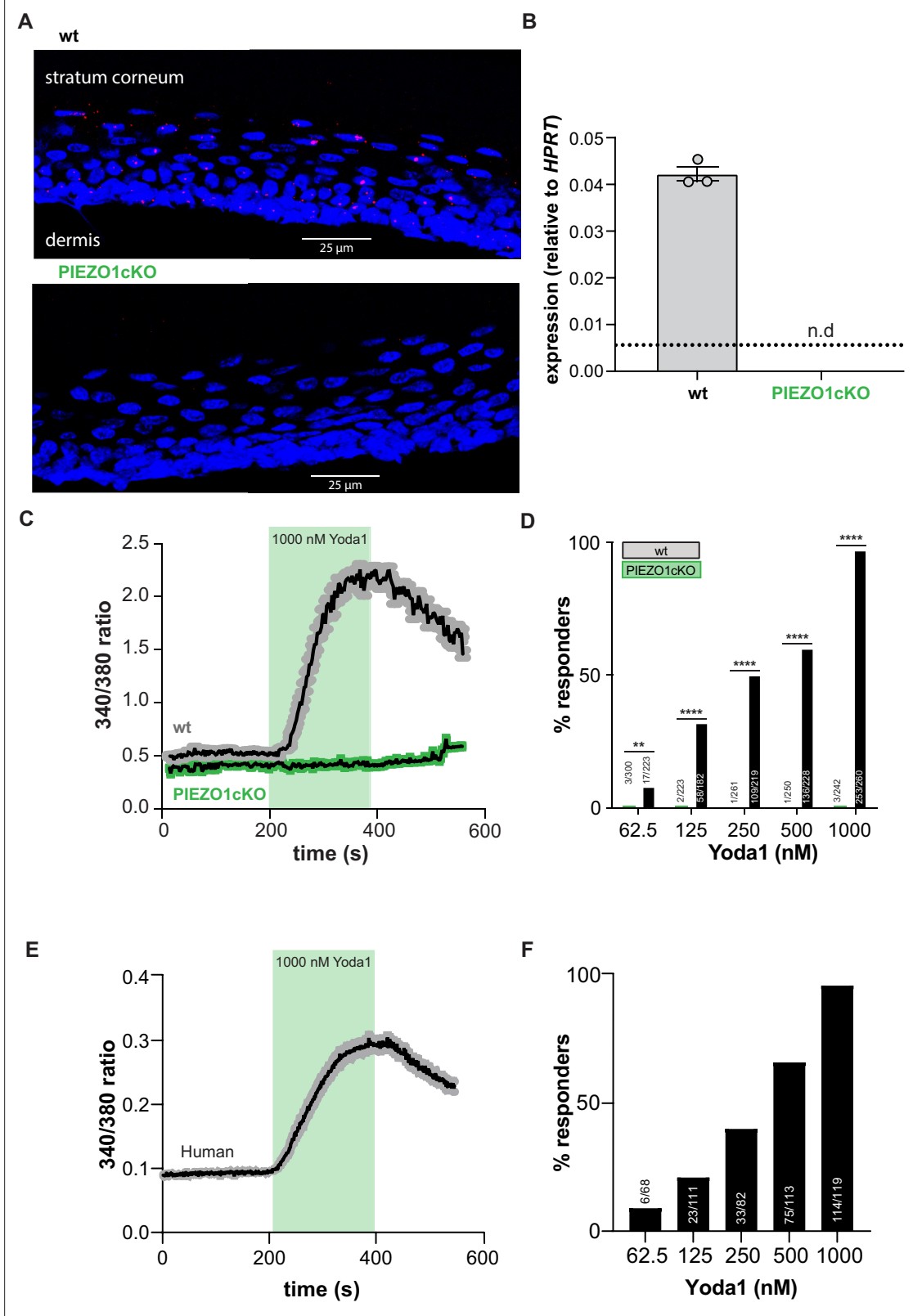

**Figure 1.** PIEZO1 is functionally expressed in mouse and human keratinocytes. (**A**) RNAscope of hindpaw glabrous skin isolated from wildtype and PIEZO1cKO mice targeting PIEZO1 mRNA (blue: DAPI, red: PIEZO1). (**B**) PIEZO1 gene expression was measured in keratinocytes isolated from wildtype (wt; n=3) and PIEZO1cKO (n=3) mice using quantitative real-time PCR. Expression levels were normalized to HPRT. Piezo1 expression was undetected in PIEZO1cKO samples. (**C**) Average calcium flux in wildtype (wt) and PIEZO1cKO keratinocytes in response to 1000 nM Yoda1; trace outline is SEM. (**D**)

*Figure 1 continued on next page*

*Figure 1 continued*

Percentage of wildtype and PIEZO1cKO keratinocytes that respond to extracellular Yoda1; cells from n=3 mice per genotype; bars are group averages; Chi square. (**E**) Calcium flux in human keratinocytes in response to 1000 nM Yoda1; trace outline is SEM. (**F**) Percentage of human keratinocytes that respond to extracellular Yoda1; cells from the skin of n=3 human donors; bars are group averages. All data are mean ± SEM unless otherwise stated. Post-hoc comparisons for all panels: **p<0.01, ****p<0.0001.

The online version of this article includes the following source data and figure supplement(s) for figure 1:

**Source data 1.** Data for panals *Figure 1B-F*.

**Figure supplement 1.** The epidermis of PIEZO1cKO animals has normal morphological features.

**Figure supplement 1—source data 1.** Individual values for epidermal thickness.

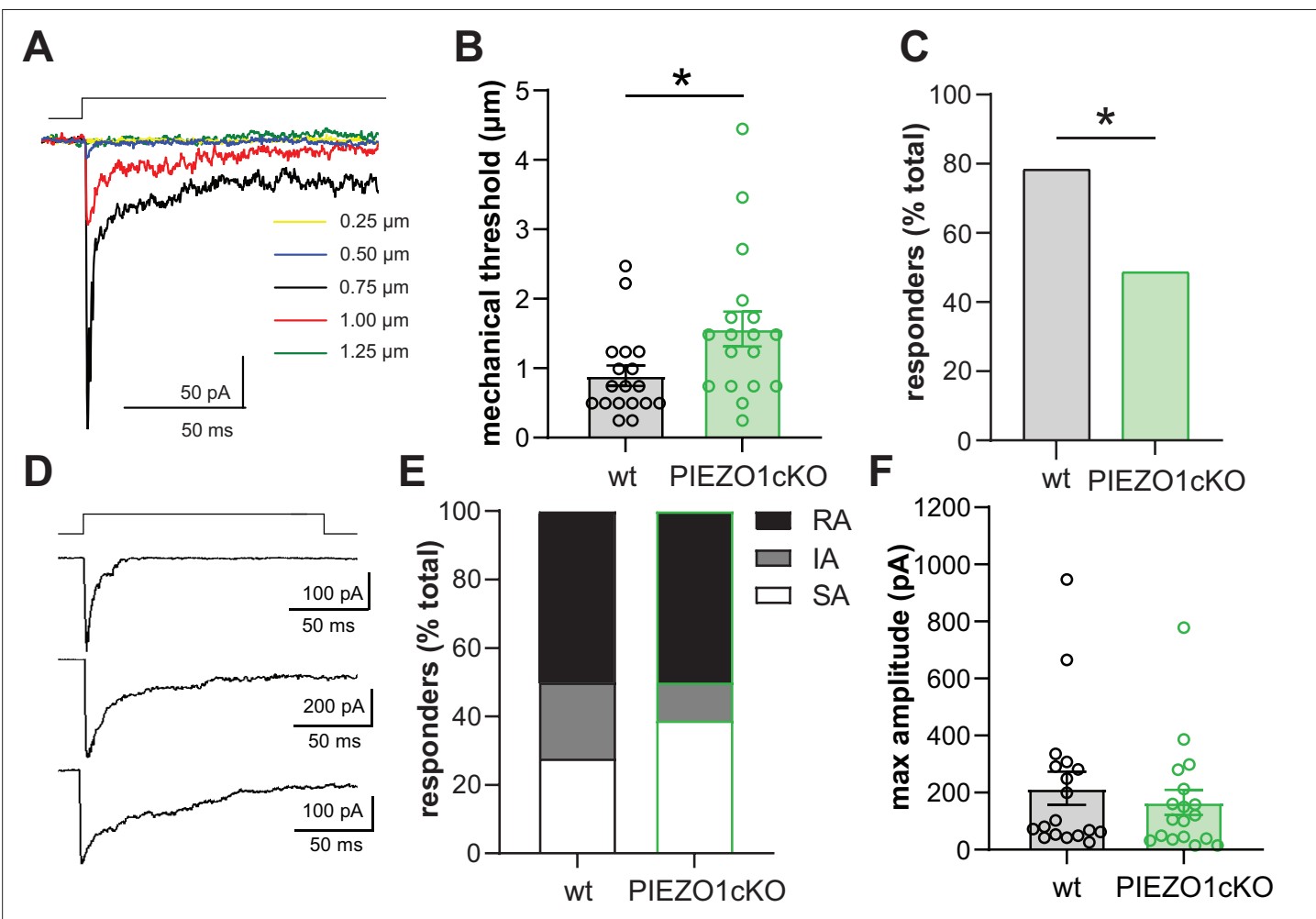

**Figure 2.** PIEZO1 deletion decreases keratinocyte mechanical sensitivity. (**A**) Examples of whole-cell recording of mechanically activated (MA) current (Vh = –40 mV) evoked in keratinocytes by a stepwise increase in membrane indentation depth. (**B**) The mechanical threshold to evoke MA current by gradual increasing of indentation depth is significantly increased in keratinocytes of PIEZO1cKO mice compared to wild-type controls; Mann-Whitney U-test. (**C**) PIEZO1 deletion significantly decreased number of keratinocytes that responded to membrane indentation with MA current (wildtype: n=23 cells; PIEZO1cKO: n=37 cells); Chi square test. (**D**) Representative traces of rapidly adapting (RA), intermediately adapting (IA), and slowly adapting (SA) MA currents induced in keratinocytes in response to membrane indentation. (**E**) Proportion of keratinocytes that responded to membrane indentation with RA, IA, and SA currents is not affected by PIEZO1 deletion (n=18 recordings of MA current per each group); Chi square and Fisher's exact post hoc test, n.s. (**F**) Maximal amplitude of MA currents in wild-type and PIEZO1cKO keratinocytes; Mann-Whitney U-test, n.s. For whole-cell patch clamp experiments, cells were harvested from n=5 mice per group. All data are mean ± SEM. *p<0.05.

The online version of this article includes the following source data for figure 2:

**Source data 1.** Data for mechanical threshold, percent responders, current profile and max amplitude of wt and PIEZO1cKO keratinocytes.

PIEZO1 is critical for setting the mechanical threshold of keratinocytes and that its deletion increases the number of mechanically insensitive cells.

## Epidermal PIEZO1 is required for normal primary afferent responses to mechanical stimulation

Since keratinocytes are known to modulate cutaneous sensory afferent responses to mechanical stimulation (*Baumbauer et al., 2015*), we next used ex vivo tibial nerve recordings to determine whether deletion of PIEZO1 in non-neuronal epidermal cells affects mechanically evoked sensory nerve firing. Aδ fibers from PIEZO1cKO mice fired fewer action potentials during mechanical stimulation of receptive fields than fibers isolated from wild-type control tissue (*Figure 3A and B*). These differences were most notable at the upper range of tested forces (100-150mN) (*Figure 3C*). In contrast, epidermal PIEZO1 knockout had no effect on the firing frequency of Aβ or C fibers (*Figure 3C–F*). No difference in mechanical thresholds between wild-type and PIEZO1cKO mice were observed for any fiber type (*Figure 3—figure supplement 1A-D*). Based on these data, the normal mechanically induced firing of Aδ primary afferent fibers depends on epidermal expression of PIEZO1.

## Activation of epidermal PIEZO1 induces paw attending responses

To determine if direct activation of epidermal PIEZO1 is sufficient to induce behavioral responses, we injected Yoda1 into the hind paw of wild-type and PIEZO1cKO mice. Yoda1 induced dose-dependent paw attending responses in wild-type mice (*Figure 4A*) but had no effect in PIEZO1cKO mice (*Figure 4B*), suggesting that the observed attending behaviors were dependent on epidermally-expressed PIEZO1. To determine if these behaviors directly result from Yoda1-induced firing of sensory neurons, we performed ex vivo teased tibial nerve recordings in tissue isolated from wild-type mice. Application of 1 mM Yoda1 failed to induce firing in any fiber type tested (Aβ n=10, Aδ n=10, C n=10 fibers, data not shown). In light of this finding, we next hypothesized that the attending behaviors observed following Yoda1 injection were due to Yoda1-induced mechanical sensitization of primary sensory afferents, such that the normally innocuous pressure of the glass floor was now sufficient to induce paw attending (*Wang et al., 2020*). In support of this hypothesis, we found that Yoda1 application increased the mechanically induced firing frequency of wild-type C fibers relative to vehicle application (*Figure 4C–F*), but had no effect on the mechanically induced firing of Aβ or Aδ fibers (10 fibers of each tested, data not shown). The increase in C fiber mechanically induced firing frequency was absent in PIEZO1cKO preparations, indicating that epidermal PIEZO1 is required for the Yoda1-induced mechanical sensitization. Furthermore, we found that intraplantar injection of 1 mM Yoda1 sensitized wild-type mice behaviors to mechanical stimulation 30 minutes following injection, an effect which was absent in the PIEZO1ckO mice (*Figure 4G*). These results indicate that Yoda1 acts at keratinocyte PIEZO1 to induce C fiber mechanical hypersensitivity that results in attending behaviors and increased mechanical sensitivity.

## Epidermal PIEZO1 mediates normal innocuous and noxious touch sensation

Finally, we investigated whether epidermal PIEZO1 is required for normal touch sensation in rodents by examining the responses of PIEZO1cKO mice and wild-type controls in a battery of behavioral assays. PIEZO1cKO mice were less sensitive to innocuous punctate stimulation with von Frey filaments (*Figure 5A–C*). Additionally, the response profiles to a dynamic light touch stimulus (paintbrush) and a noxious punctate (needle) stimulus were altered in PIEZO1cKO mice; PIEZO1cKO mice responded less frequently to both paintbrush and needle stimulation than wild-type controls (*Figure 5D and E*). PIEZO1cKO mice did not exhibit general somatosensory deficits, as animal responses to heat (*Figure 5F*) and cold (*Figure 5G*) were identical to those observed in wild-type mice. Furthermore, we utilized high-speed videography to capture sub-second behavioral features in response to a single hind paw application of von Frey filaments (0.4 g, 1.4 g, 4 g), a paintbrush, or a needle. Both reflexive (paw withdrawal height and velocity) and affective (pain score) behaviors were measured (*Abdus-Saboor et al., 2019*; *Jones et al., 2020*). PIEZO1cKO mice were less sensitive to stimulation with von Frey filaments; fewer PIEZO1cKO animals responded to the 0.6 g filament (*Figure 5—figure supplement 1A*). Raw values for reflexive and affective responses (*Figure 5—figure supplement 1B-D*) of each animal were converted to normalized z-scores to generate a cumulative sensitivity score for

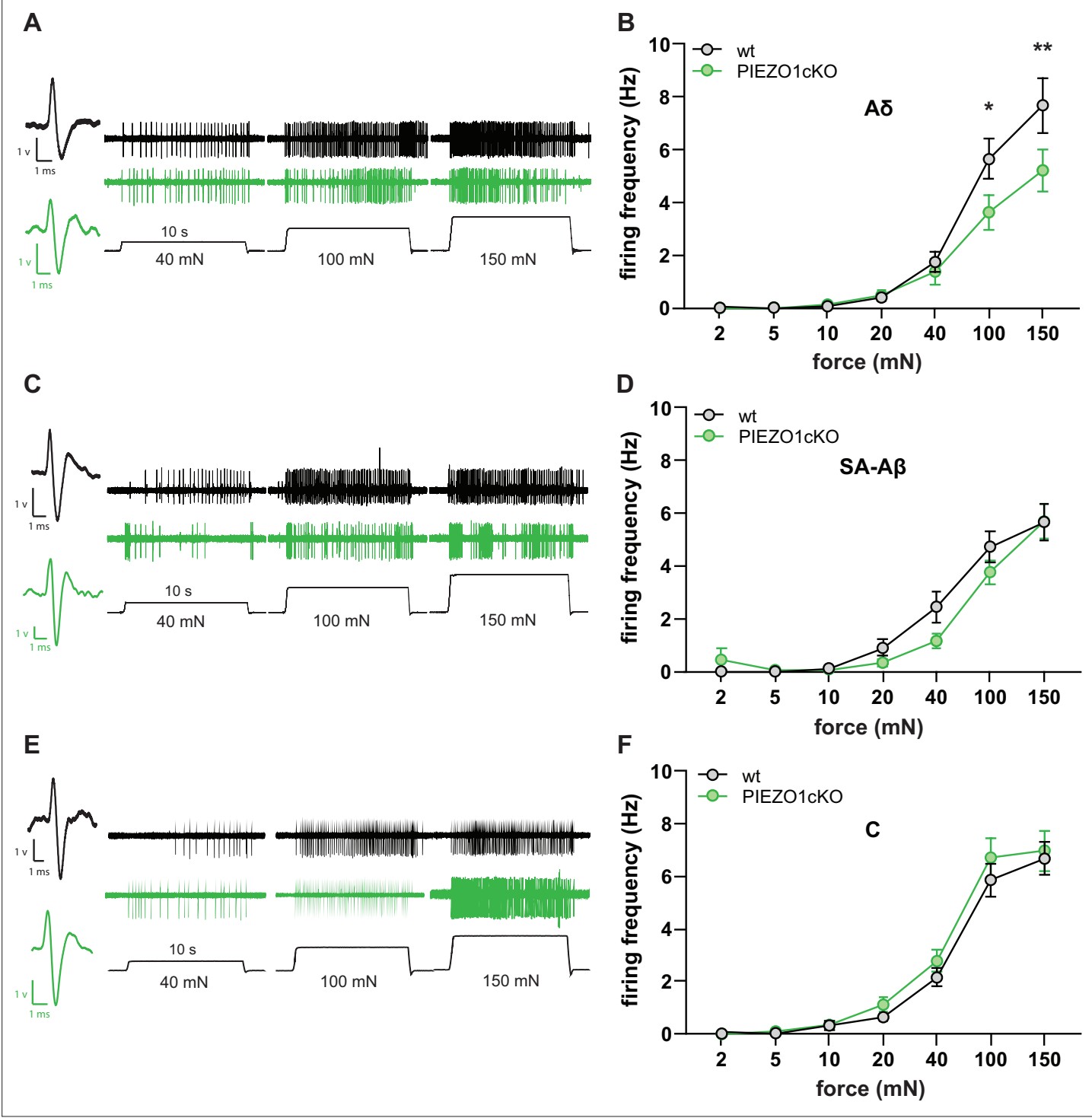

**Figure 3.** Normal mechanically-induced primary afferent firing requires epidermal PIEZO1 expression. Ex vivo tibial nerve recordings of Piezo1cKO and wildtype (wt) mice. (**A**) Aδ fiber example traces. (**B**) Mean mechanically induced firing rates of Aδ fibers (n=33 wt and 30 PIEZO1cKO fibers). (**C**) SA-Aβ fiber example traces. (**D**) Mean mechanically induced firing rates of SA-β fibers (n=33 wt and 28 PIEZO1cKO fibers). (**E**) C fiber example traces. (**F**) Mean mechanically induced firing rates of C fibers (n=30 wt and 34 PIEZO1cKO fibers). For all recordings, the mechanical stimulus was applied to the skin for 10 seconds. All data are mean ± SEM; 2-way ANOVA and Sidak post-hoc comparisons for firing frequency panels: *p<0.05, **p<0.01; fibers from n=17–19 mice.

The online version of this article includes the following source data and figure supplement(s) for figure 3:

**Source data 1.** Data for mechanically induced firing frequency of sensory afferents.

*Figure 3 continued on next page*

*Figure 3 continued*

**Figure supplement 1.** PIEZO1 deletion does not alter sensory fiber mechanical thresholds.

**Figure supplement 1—source data 1.** Data for mechanical threshold.

each stimulus (*Figure 5H*); these cumulative scores were then averaged into a combined mechanical sensitivity score for each animal (*Figure 5I*). PIEZO1cKO mice exhibited decreased mechanical sensitivity scores compared to wild-type animals, an effect that was most apparent in the needle stimulus (*Figure 5H and I*).

## Discussion

Until very recently, it was assumed that sensory neurons were the primary, and in some cases, sole transducers of innocuous and noxious stimuli in skin (*Moehring et al., 2018b*; *Talagas et al., 2020b*; *Hill and Bautista, 2020*). However, this dogma has essentially been negated by recent work that demonstrates how non-neuronal cells, including keratinocytes, are required for the normal detection and coding of somatosensory stimuli in the peripheral nervous system (*Maksimovic et al., 2014*; *Abdo et al., 2019*; *Moehring et al., 2018a*; *Neubarth et al., 2020*; *Sadler et al., 2020*). Here, we show for the first time that PIEZO1 is one of the critical mechanotransducers in keratinocytes that enables these cells to encode mechanical force and convey this signal to sensory afferent terminals.

### Epidermal PIEZO1 is critical for normal gentle and noxious touch detection

Our findings are the first to demonstrate a role for PIEZO1 in tactile sensation. These data complement previous work showing the necessity of this channel as a sensor of mechanical forces in the lung, bladder, and circulatory system (*Li et al., 2014*; *Ranade et al., 2014a*; *Friedrich et al., 2019*; *Dalghi et al., 2019*), and the necessity of family member PIEZO2 in the detection of light touch (*Maksimovic et al., 2014*; *Woo et al., 2014*; *Hoffman et al., 2018*; *Ranade et al., 2014b*; *Chesler et al., 2016*). Furthermore, these results indicate that the functional contributions of PIEZO1 activity in the epidermis are multifaceted, as PIEZO1 also regulates epidermal cell extrusion and wound healing (*Eisenhoffer et al., 2012*; *Holt et al., 2020*). Epidermal PIEZO1 deletion decreased animal behavioral responsiveness to a range of intensities and qualities of mechanical stimuli. This contrasts to the behaviors observed when PIEZO2 is deleted from various peripheral cell types; animals are not able to detect very light punctate stimuli when PIEZO2 is deleted from Merkel cells (*Maksimovic et al., 2014*; *Woo et al., 2014*), and similarly, deletion of PIEZO2 from dorsal root ganglion neurons results in behavioral deficits to light punctate and dynamic stimuli but not to stimuli in the high to noxious range of forces (*Ranade et al., 2014b*). While Merkel cell and neuronal PIEZO2 specifically mediate sensitivity to light touch, epidermal PIEZO1 appears to be a more general amplifier of cutaneous mechanical stimuli. It is important to note that sensory neurons are capable of detecting and encoding aspects of mechanical stimuli without input from epidermal cells; neither epidermal PIEZO1 deletion nor optogenetic inhibition of keratinocytes completely abolishes touch sensation (*Baumbauer et al., 2015*; *Moehring et al., 2018a*), but rather both manipulations decrease neuronal and behavioral mechanical sensitivity. Keratinocyte activation and subsequent signaling appears to function in concert with sensory neurons and other cutaneous end organ structures to amplify normal touch sensation. Although it is possible that the mechanical deficits displayed by the PIEZO1cKO mice were due to indirect developmental effects of PIEZO1 deletion on the structure or function of the epidermis rather than a decrease in keratinocyte mechanical signaling, we observed no deficits in cold or heat behaviors in the PIEZO1cKO mice, suggesting that general somatosensation was not affected in the mutants. Furthermore, we did not observe gross changes in epidermal morphology, suggesting that general epidermal disorganization was not the main driver of the decreased mechanical sensitivity exhibited by the PIEZO1cKO mice.

### PIEZO1 is a key keratinocyte mechanotransducer

Previous work has demonstrated that keratinocytes are inherently mechanically sensitive (*Tsutsumi et al., 2009*; *Goto et al., 2010*; *Koizumi et al., 2004*). However, the molecular transducer that converts force into cellular responses in keratinocytes was unknown (*Moehring et al., 2018b*; *Talagas*

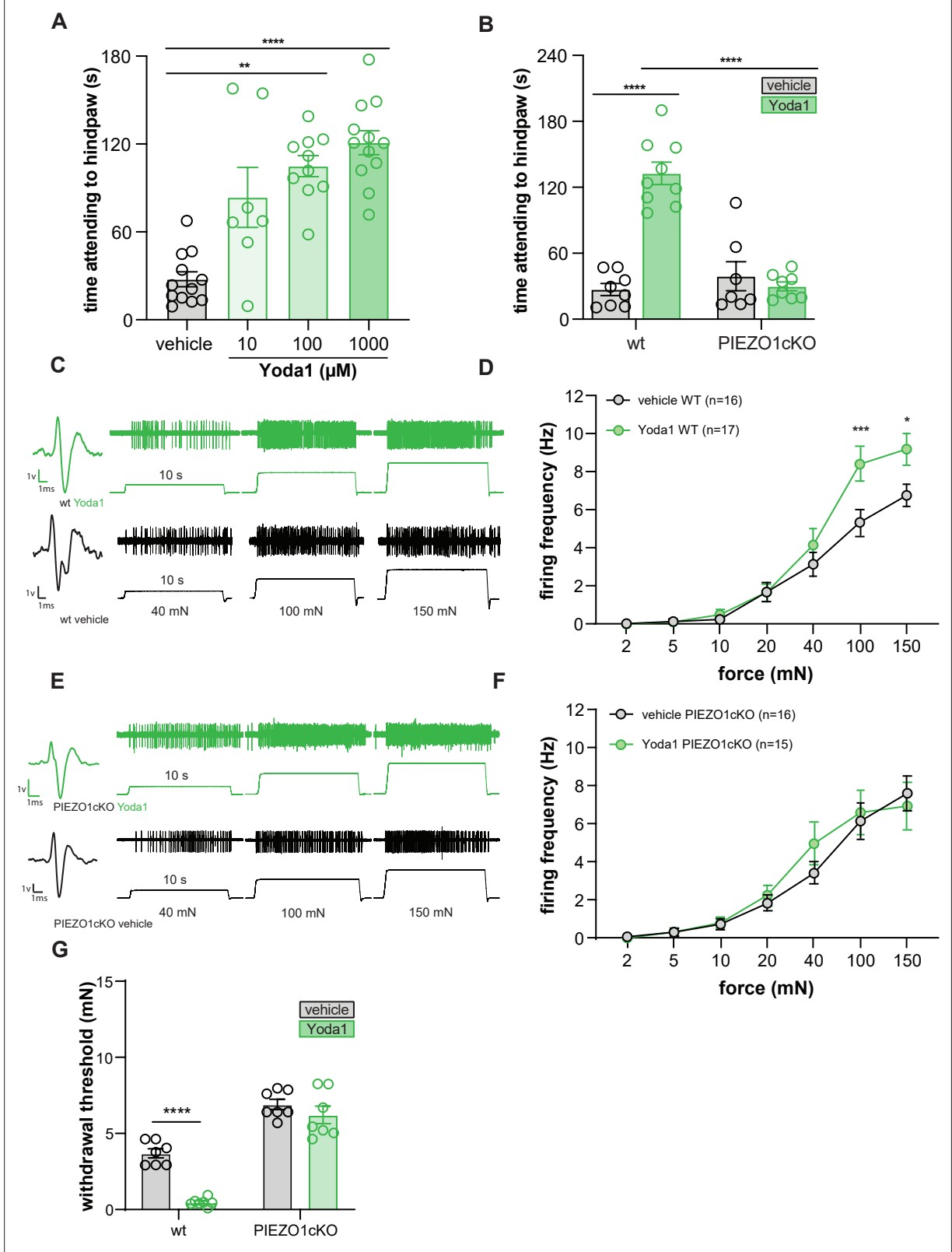

**Figure 4.** Yoda1 induces paw attending behaviors and C fiber mechanical hypersensitivity. (**A**) Yoda1-induced attending behaviors in wildtype mice; Kruskal Wallis test. (**B**) 1 mM Yoda1 induced attending behaviors in wildtype and PIEZO1cKO mice; 2-way ANOVA. (**C**) Example traces from wildtype preparations exposed to 1 mM Yoda1 or vehicle during mechanical testing. (**D**) Mean mechanical firing frequency of C fibers from wildtype animals exposed to 1 mM Yoda1 or vehicle. (**E**) Example traces of PIEZO1cKO preparations exposed to 1 mM Yoda1 or vehicle during mechanical testing.

*Figure 4 continued on next page*

*Figure 4 continued*

(**F**) Mean mechanical firing frequency of C fibers from PIEZO1cKO animals exposed to 1 mM Yoda1 or vehicle. Fibers from n=12–14 mice. For all teased fiber recordings, the mechanical stimulus was applied to the skin for 10 s; 2-way ANOVA. (**G**) Von Frey mechanical thresholds of wildtype and PIEZO1cKO mice tested 30 min after an injection of 1 mM Yoda1 or vehicle; 2-way ANOVA . All data are mean ± SEM unless otherwise stated. Post-hoc comparisons for all panels: **p<0.01, ***p<0.001, ****p<0.0001.

The online version of this article includes the following source data for figure 4:

**Source data 1.** Data for time attending to hindpaws, Yoda1 induced firing frequency, and Yoda1 induced mechanical hypersensitivity.

*et al., 2020b*). Here, we show that PIEZO1 deletion substantially reduces the number of keratinocytes that respond to membrane indentation; 51.35% of keratinocytes were insensitive to mechanical indentation following deletion of PIEZO1. These results indicate that PIEZO1 is a key mechanotransducer in keratinocytes, mirroring the role PIEZO2 plays in both Merkel cells and dorsal root ganglia (DRG) neurons (*Woo et al., 2014*; *Ranade et al., 2014b*; *Coste et al., 2010*). In the population of PIEZO1cKO keratinocytes that retained mechanical sensitivity, the mechanical threshold was higher than those observed in wild-type keratinocytes. Furthermore, PIEZO1 deletion reduced but did not eliminate the number of rapidly and intermediately adapting currents. These changes in mechanical response properties suggest that one or more unknown mechanotransducers function to detect mechanical stimuli in a subset of keratinocytes. One potential candidate for this function is PIEZO2, which is shown to mediate the rapidly adapting mechanical currents in DRG neurons and Merkel cells. However, keratinocytes express low levels of PIEZO2 transcript, making it unlikely to be the primary contributor to the remaining keratinocyte mechanical sensitivity (*Hoffman et al., 2018*; *Coste et al., 2010*). In addition to PIEZO1, keratinocytes express a host of ion channels that may modulate keratinocyte mechanical responses downstream of bona fide mechanotransducers. These include members of the transient receptor potential (TRP) family of ion channels, such as TRPV4 and TRPC5 (*Peier et al., 2002*; *Tu et al., 2005*; *O'Neil and Heller, 2005*; *Shen et al., 2015*). Interestingly, TRPV4 expression in keratinocytes is required for the development of mechanical allodynia in a mouse model of sunburn pain (*Moore et al., 2013*). Additionally, TRPC5 expression is required for the development of mechanical allodynia in several inflammatory and neuropathic pain models (*Sadler et al., 2021*), although the contribution of epidermal TRPC5 to injury induced mechanical hypersensitivity remains to be explored. Whether TRPV4, TRPC5, and/or other channels contribute to normal keratinocyte mechanotransduction, or the potential sensitization of keratinocyte mechanotransduction following injury, warrants further investigation beyond this current study.

## Epidermal PIEZO1 is important for mechanically induced sensory afferent firing

Our teased fiber recordings revealed that deletion of PIEZO1 from the epidermis selectively decreased the firing frequency of Aδ fibers. This difference was most notable at the higher range of forces tested, indicating that epidermal PIEZO1 is important for the high intensity firing of Aδ fibers. However, the PIEZO1cKO animals had deficits in behavioral responses to both light touch and high-threshold mechanical stimuli. A potential explanation is that the reflexive behavioral responses to touch may rely on the summation of activity in many overlapping receptive fields, and therefore, activity in keratinocytes from a broad area of skin. Alternatively, the mechanical responsiveness of a single afferent fiber may rely on signaling from far fewer keratinocytes. Interestingly, Aδ nociceptors have been shown to mediate behavioral responses to pinprick stimuli (*Arcourt et al., 2017*), which was the behavior most affected by loss of epidermal PIEZO1 in our high-speed imaging experiments.

We found that epidermal PIEZO1 deletion had no effect on the mechanical responses of SA-Aβ fibers. This is likely because Merkel cells tune the mechanical responses of these fibers and Merkel cell mechanical sensitivity is primarily mediated by PIEZO2 (*Maksimovic et al., 2014*; *Woo et al., 2014*). Surprising to us, however, was that C fiber afferents from PIEZO1cKO and wild-type preparations exhibited similar mechanical firing frequency and mechanical thresholds. This was unexpected since many C fiber terminals are closely apposed to keratinocytes (*Zylka et al., 2005*) and, in experiments performed by Baumbauer, Deberry, and Adelman, et al., optogenetic inhibition of epidermal cells decreased C fiber mechanical firing in 12 of 25 fibers tested (*Baumbauer et al., 2015*). It is possible that the absence of an effect of keratinocyte PIEZO1 deletion on C fiber mechanical sensitivity may be

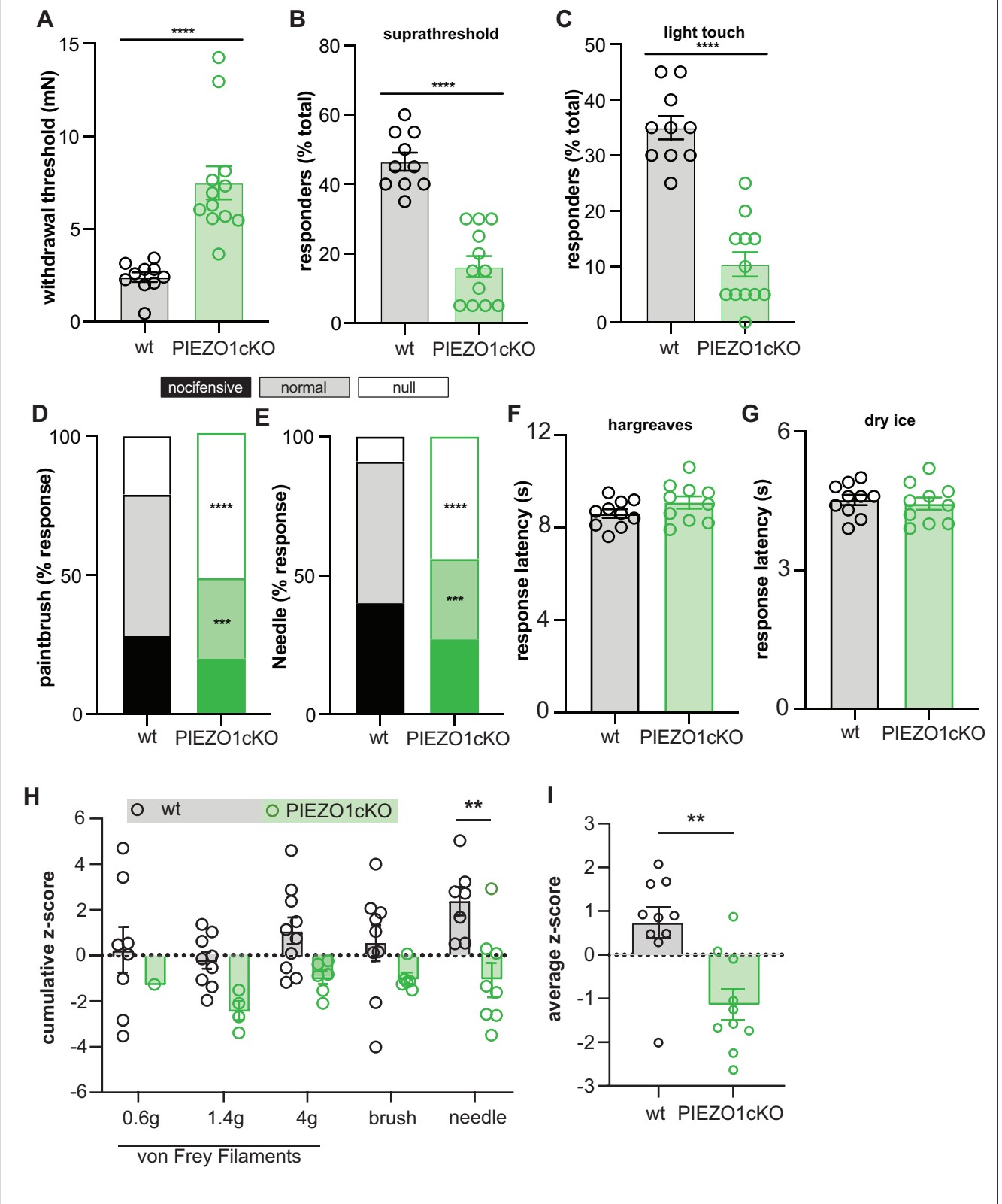

**Figure 5.** Epidermal Piezo1 is required for normal innocuous and noxious mechanosensation. (**A**) Von Frey mechanical thresholds of wildtype and PIEZO1cKO mice; Mann-Whitney U-test. (**B**) Wildtype and PIEZO1cKO responses to repeated suprathreshold (3.61 mN) von Frey filament stimulation; Mann-Whitney U-test. (**C**) Wildtype and PIEZO1cKO responses to repeated static light touch (0.6 mN von Frey filament) stimulation; Mann-Whitney U-test. (**D**) Response characterization to paintbrush swiping across hindpaw; n=10–12; bars are group averages; Chi Square test. (**E**) Response

*Figure 5 continued on next page*

*Figure 5 continued*

characterization to noxious needle hindpaw stimulation; n=10–12; bars are group averages; Chi Square test. (**F**) Withdrawal latency to radiant heat hindpaw stimulation; Student's (two-tailed) *t* test, n.s. (**G**) Withdrawal latency to dry ice hindpaw stimulation; Student's (two-tailed) *t* test, n.s. (**H**) High-speed imaging mechanical sensitivity scores in response to von Frey (0.6 g, 1.4 g, 4 g), brush, and needle stimulation in wildtype and PIEZO1cKO mice; two-way ANOVA . Cumulative z-scores were calculated from paw height, paw velocity, and pain score at each stimulus. (**I**) Average high-speed imaging mechanical sensitivity score across all stimuli for each animal. All data are mean ± SEM unless otherwise stated. Post-hoc comparisons for all panels: \*\*p<0.01, \*\*\*p<0.001, \*\*\*\*p<0.0001.

The online version of this article includes the following source data and figure supplement(s) for figure 5:

**Source data 1.** Data for behavioral mechanical sensitivity.

**Figure supplement 1.** High-speed imaging of PIEZO1cKO and wildtype mice.

**Figure supplement 1—source data 1.** Data for high speed imaging of mechanical sensitivity.

---

explained by differences in our teased fiber recording methods compared to those used by Baumbauer and colleagues; we apply our mechanical stimulus to the dermal side of the skin, whereas Baumbauer et al. applied their mechanical stimulus to the corneum (i.e., how an external mechanical stimulus would naturally be applied to the skin in vivo). Alternatively, because many C fibers terminate more superficially in the epidermis than Aδ fibers (*Zylka et al., 2005*), it is possible that the differentiated keratinocytes of the outer epidermis rely on a different mechanotransducer than PIEZO1 to encode tactile information. We recently reported that keratinocytes release ATP in response to mechanical stimulation, which subsequently acts at purinergic receptors on sensory terminals to mediate tactile sensation (*Moehring et al., 2018a*). Therefore, ATP may be one of the signaling molecules linking epidermal PIEZO1 activity to sensory neuron responses. In addition to ATP, keratinocytes can release a variety of other neuroactive factors, including calcitonin gene-related peptide β, acetylcholine, glutamate, epinephrine, neurotrophic growth factors, endothelin-1, and cytokines (*Hou et al., 2011*; *Shi et al., 2013*; *Barr et al., 2013*; *Moore et al., 2013*; *Lumpkin and Caterina, 2007*) the contribution of these ligands to neuro-epithelial mechanical signaling remains to be explored.

An important caveat to our findings is that the use of the *keratin 14* (K14) promotor to target PIEZO1 deletion in the epidermis means that we cannot definitively rule out the contribution of other K14-expressing cells to our experiments. For example, Merkel cells express K14 and would have PIEZO1 deleted from them (*Maksimovic et al., 2014*; *Woo et al., 2014*). However, given that keratinocytes make up the vast majority of cells in the epidermis (>95%) (*Fuchs, 1995*), and Merkel cells express minimal PIEZO1 (*Maksimovic et al., 2014*; *Hoffman et al., 2018*), we hypothesize that our findings are largely mediated by keratinocytes. Furthermore, it is unlikely that our PIEZO1 deletion is targeting the recently discovered sensory Schwann cells, as these cells do not express Keratin 14 (P. Ernfors, personal communication, unpublished data).

## Chemical activation of epidermal PIEZO1 induces behavioral mechanical hypersensitivity

Yoda1 application induced robust calcium flux in both isolated mouse and human keratinocytes. The functional expression of PIEZO1 in human keratinocytes is intriguing as it suggests that epidermal PIEZO1 may also play a role in human touch sensation. PIEZO1 loss of function mutations have been identified in human patients (*Lukacs et al., 2015*; *Alper, 2017*), but to our knowledge, quantitative sensory testing, like that which has been performed in patients with PIEZO2 loss of function mutations (*Chesler et al., 2016*), has not yet been completed in these individuals. This type of study could reveal important information about the role of PIEZO1 in human touch sensation.

We found that intraplantar injection of the PIEZO1 specific agonist Yoda1 induced paw attending behaviors in wild-type but not PIEZO1cKO mice, suggesting that activation of epidermal PIEZO1 is sufficient to induce behavioral responses. However, application of Yoda1 to the receptive fields of functionally identified primary afferent fibers failed to induce action potential firing. This result was surprising given that Yoda1 induced calcium responses in isolated keratinocytes and PIEZO1 is functionally expressed in itch-specific sensory neurons (*Hill et al., 2022*). Because we applied Yoda1 to the dermal layer of the skin, it is possible that Yoda1 failed to penetrate sufficiently to the PIEZO1 expressing cells in the epidermis to induce sensory fiber firing. Interestingly, we observed an increase in C fiber mechanical sensitivity following Yoda1 application, an effect that was dependent on

epidermal PIEZO1 expression. Therefore, an alternative explanation is that the Yoda1 induced behaviors may reflect increased hindpaw mechanical sensitivity, such that the innocuous force produced by the paw resting on the glass floor becomes sufficient to induce paw attending. In line with this hypothesis, we found that intraplantar injections of Yoda1 induced mechanical allodynia in wild-type but not PIEZO1cKO mice. Indeed, while Yoda1 can activate PIEZO1 channels on its own, it prominently sensitizes PIEZO1 to mechanical stimulation (*Syeda et al., 2015*; *Lacroix et al., 2018*). Such an increase in PIEZO1 mechanical sensitivity may explain why the effects of Yoda1 on sensory fiber firing were only observed in the presence of mechanical force. Interestingly, the team of Baumbauer, DeBerry, and Adelman reported that subthreshold mechanical stimulation could induce firing in high-threshold mechanically sensitive afferents when paired with optogenetic activation of keratinocytes (*Baumbauer et al., 2015*). This suggests that sensitization of keratinocyte mechanical signaling may enhance sensory afferent responses to force. This idea is intriguing, as injury induced sensitization of keratinocyte mechanotransduction may contribute to the development of mechanical allodynia and hyperalgesia. Alterations in keratinocyte function and signaling contribute to inflammatory and neuropathic cutaneous pain states, such as psoriasis, dermatitis, fibromyalgia, complex regional pain syndrome, and postherpetic neuralgia (*Benhadou et al., 2019*; *Kim and Leung, 2018*; *Li et al., 2009*; *Evdokimov et al., 2020*; *Zhao et al., 2008*). Furthermore, in a mouse model of sunburn pain, UVB light exposure resulted in profound mechanical and heat allodynia, effects which were completely dependent on UVB-induced activation and sensitization of keratinocyte expressed TRPV4 (*Moore et al., 2013*). Thus, injury-induced sensitization of keratinocyte mechanotransducers may enhance normal keratinocyte activation in response to mechanical stimulation and subsequent signaling to sensory neurons.

## Conclusion

In summary, the data presented here demonstrate that epidermal PIEZO1 is critical for normal touch sensation in mice and that PIEZO1 is also expressed and functional in human keratinocytes. Future studies will focus on whether PIEZO1 signaling and resulting keratinocyte activity may be altered in injury models leading to mechanical hypersensitivity and allodynia.

# Materials and methods

## Animals

To target epidermal keratinocytes, a *Keratin14* (*Krt14*)$^{Cre}$ driver was used, as *Krt14* is expressed in all keratinocytes as early as E9.5 (*Byrne et al., 1994*; *Wang et al., 1997*; *Dassule et al., 2000*). These mice (Jackson Laboratory, Farmington) were mated with *Piezo1*$^{loxp/loxp}$ animals (Jackson Laboratory) to produce offspring that lacked PIEZO1 in K14-expressing cells and were genotyped as either *Krt14Cre$^+$ Piezo1*$^{loxp/loxp}$ (PIEZO1cKO) or *Krt14Cre$^-$ Piezo1*$^{loxp/loxp}$ (wild type; wt). For all studies a mixture of male and female mice aged 6–20 weeks were used. Male and female mice were analyzed separately, and no sex differences were observed. Therefore, data shown in graphs show combined results of both sexes.

Animals had ad libitum access to food and water and were housed in a climate-controlled room with a 12:12 light:dark cycle, on Sani-Chips aspen wood chip bedding (P.J. Murphy Forest and Products, New Jersey) with a single pack of ENVIROPAK nesting material (W.F Fisher & Son, Inc, New Jersey). All animals were group housed with a minimum of 3 mice per cage. All animal procedures were strictly adhered according to the NIH Guide for the Care and Use of Laboratory animals and were performed in accordance with the Institutional Animal Care and Use Committee at the Medical College of Wisconsin (approval #0383). This manuscript adheres to the applicable ARRIVE guidelines.

## Primary keratinocyte cell culture

Primary mouse keratinocytes were cultured from glabrous hindpaw tissue as previously described (*Moehring et al., 2018a*; *Sadler et al., 2020* ). Briefly, isolated glabrous hindpaw skin was incubated at room temperature (RT) in 10 mg/mL dispase (Gibco, ThermoFisher Scientific, Waltham, MA) for 45 min. Primary human keratinocytes were isolated from human skin tissue (procured through the MCW Tissue Bank) as previously described (*Sadler et al., 2020*). Human skin was incubated overnight at 4 ° C in 10 mg/mL dispase. Following the dispase incubation, the epidermal sheet was separated from the dermis and incubated at RT in 50% EDTA (Sigma-Aldrich) and 0.05% trypsin (Sigma) in Hanks'

Balanced Salt Solution without calcium chloride, magnesium chloride and magnesium sulfate (Gibco) for 27 min. After 27 min, 15% heat inactivated fetal bovine serum (ThermoFisher Scientific, Carlsbad, CA) was added and the epidermal sheets were rubbed against the base of a petri dish to separate the keratinocytes. Keratinocytes were grown for 3 days in Epilife media (Gibco) supplemented with 1% human keratinocyte growth supplement (Gibco), 0.2% GibcoAmphotericin B (250 µg/mL of Amphotericin B and 205 µg/mL sodium deoxycholate, Gibco) and 0.25% penicillin-streptomycin (Gibco) on laminin coated coverslips. Plates were kept at 37 °C and 5% $CO_2$ conditions. Growth media was exchanged every 2 days.

## RNA isolation and quantitative real-time PCR

Keratinocytes were isolated from the glabrous hindpaw skin of PIEZO1cKO animals and littermate controls as described above. RNA was isolated from these keratinocytes using the PureLink RNA Mini Kit (LifeTechnologies, Carlsbad, CA). Total RNA content was assessed with a Nanodrop Lite spectrophotometer (Thermo Scientific, Wilmington, DE). cDNA was generated using the SuperScript III First-Strand Synthesis System (Invitrogen, LifeTechnologies). Quantitative real-time PCR reaction was performed on a BIO-RAD CFX96 system (Bio-Ra, Hercules, CA). Samples were run in triplicate. Gene expression was normalized to *Hprt*. The following primer sets were used: *mPiezo1*-qF: CTTACACG GTTGCTGGTTGG; *mPiezo1*-qR: CACTTGATGAGGGCGGAAT; *Hprt*-qF: GTTAAGCAGTACAGCC CCAAA; *Hprt*-qR: AGGGCATATCCAACAACAAACTT (*Wang et al., 2020*).

## Calcium imaging

Calcium imaging was performed on keratinocytes on their third day in culture. Keratinocytes were loaded with 2.5 µg/mL Fura-2-AM, a dual-wavelength ratiometric calcium indicator dye, in 2% BSA for 45 min at RT then washed with extracellular buffer for 30 min. Keratinocytes were superfused with RT extracellular buffer (pH 7.4, 320 Osm) containing (in mM) 150 NaCl, 10 HEPES, 8 glucose, 5.6 KCl, 2 $CaCl_2$, and 1 $MgCl_2$ and viewed on a Nikon Eclipse TE200 inverted microscope. Nikon elements software (Nikon Instruments, Melville, NY) was used to capture fluorescence images at 340 and 380 nm. Responsive cells were those that exhibited >30% increase in 340/380 nm ratio from baseline. Yoda-1 (Sigma-Aldrich) was prepared from a 10 mM stock solution (5 mg Yoda1 in DMSO) at 62.5, 125, 250, 500, and 1000 (nM) concentrations in extracellular buffer for the dose response curve. Yoda-1 was applied for 3 min and washed out for 3 min.

## Behavioral assays

For all spontaneous and evoked behavior experiments, the experimenter was blinded to genotype throughout testing and data entry. Animals were tested between 8am and 1pm and were allowed to acclimate for at least an hour to the new surroundings and experimenter before any behavior testing was performed.

Mechanical sensitivity: A battery of different assays using various stimuli were utilized to determine the mechanical sensitivity of the of PIEZO1cKO and wild-type littermate controls. Using the Up-Down method and a series of calibrated von Frey filaments ranging from 0.20 to 13.73 mN, mechanical thresholds of the glabrous hindpaw skin were assessed as previously described (*Chaplan et al., 1994*; *Dixon, 1980*). Additionally, the hindpaw skin was stimulated 10 times using a 3.61 mN von Frey Filament in the suprathreshold assay, and using a 0.6 mN von Frey Filament in the static light touch assay. The number of stimulus-evoked paw withdrawals were recorded (*Weyer et al., 2016*). Furthermore, we utilized a paintbrush that was stroked 10 times across the hindpaw and responses were categorized as: normal/innocuous (simple withdrawal of the paw), noxious (elevating the paw for extended periods of time, flicking and licking of the paw), and null responses (no withdrawal) (*Cowie et al., 2019*). Lastly, noxious mechanical sensitivity was assessed using the needle assay (*Hogan et al., 2004*; *Garrison et al., 2014*), where responses were categorized similar to the paintbrush assay.

To test heat sensitivity, mice were placed in a small plexiglass enclosures on top of a glass plate, and a focal radiant heat source was applied to the plantar hindpaw. The response latency to hind paw withdrawal from the heat stimulus was quantified (*Hargreaves et al., 1988*), with a cut off at 25 s to avoid tissue damage. Each paw was tested 4 times, with 5 min of rest between testing, and results were averaged for each animal.

To test cold sensitivity, animals were placed in small plexiglass enclosures on top of a thin 2.5-mm-thick glass plate, and powdered dry ice packed into a 10 mL syringe with the top cut off was pressed against the glass beneath the plantar surface of the hindpaw (*Brenner et al., 2012*). Withdrawal latencies were recorded three times for each paw, with 5 min of break in between testing, and results were averaged for each animal. The maximum time allowed for withdrawal was 20 s to avoid potential tissue damage.

Spontaneous behaviors were recorded in response to intraplantar Yoda-1 injections (10 µM, 100 µM, and 1 mM). Yoda1 or vehicle was injected into the plantar surface of the hindpaw and behaviors were recorded for 10 min following the injection. Videos were analyzed offline by an experimenter blinded to both genotype and treatment. Behaviors exhibited by the animals included biting and licking of the hindpaw.

High speed imaging was used to record withdrawal behaviors in response to von Frey filaments (0.6 g, 1.4 g, 4 g), paintbrush, and needle stimulation. Videos were recorded on a FastCAM UX100 high-speed camera (Photron, Tokyo, Japan) for five seconds at 2000 frames per second starting with the application of the mechanical stimulus to the hindpaw. Videos were analyzed offline using Fastcam software (Photron). Paw height was measured as the distance from the apex of the first upward movement to the point directly below it on the mesh, as previously described. Paw velocity was measured by taking two points at different frames during the first upward movement of the paw. Pain score was measured based on the presence of jumping, paw shaking, or paw guarding behaviors. The presence of each of these behaviors was worth 1 point toward the pain score. An animal displaying 2 of the 3 behaviors would receive a score of 2, whereas an animal exhibiting all three behaviors in response to the stimulus would receive a score of 3 (*Abdus-Saboor et al., 2019*; *Jones et al., 2020*).

## H&E staining

A hematoxylin and eosin stain was performed to assess the general morphology of the glabrous skin of PIEZO1cKO and wild-type littermate controls. Glabrous skin of PIEZO1cKO and wild-type animals was dissected, and the tissue was fixed in 4% formaldehyde. The skin was processed, embedded in paraffin, sectioned into 4 µm sections and dried at RT until subsequent staining at the MCW Histology Core. Rehydrated sections were stained in hematoxylin for 3 min, washed in Richard-Allan Scientific Signature Series Clarifier 1,2 (for 45 sec, dipped for 30 sec in 0.1% ammonia water (bluing agent)), stained in eosin for 30 s, washed four times using 100% EtOH and lastly rinsed in Xylene. Slides were scanned using a Hamamatsu Nanozoomer HT slide scanner (Hamamatsu Photonics, K.K., Hamamatsu City, Japan) and images were assessed using NDP.View 2 software (Hamamatsu Photonics).

## Patch clamp recordings

On the first day of culture, keratinocyte recordings were performed. Keratinocytes were superfused with RT extracellular normal HEPES solution containing (in mM): 127 NaCl, 3 KCl, 2.5 CaCl2,1 MgCl2, 10 HEPES, and 10 glucose, pH 7.35±0.05, and viewed on a Nikon Eclipse TE200 inverted microscope. Keratinocytes were patch clamped in voltage clamp mode (holding voltage –40 mV) with a borosilicate glass pipette (Sutter Instrument Company, Novato, CA) filled with intracellular solution containing (in mM): 135 KCl, 10 NaCl, 1 MgCl2, 1 EGTA, 0.2 NaGTP, 2.5 ATPNa2, 10 glucose and 10 HEPES, pH 7.25±0.05. Mechanical stimulation was elicited using a glass rod positioned approximately 2 µm from keratinocyte's membrane and driven by a piezo stack actuator (PA25, PiezoSystem Jena, Jena, Germany) at a speed of 39.17 µm/ms. Keratinocytes were stimulated with a series of mechanical steps in 0.25 µm increments applied for 150ms every 30 s to avoid sensitization/desensitization of the cell membrane. Data was recorded using (Axon pCLAMP 11 software using Digidata 1550B and Axopatch 200B amplifier Molecular Devices LLC, San Jose, CA). The cell was considered mechanically insensitive if the inward current was not observed with 5.0 µm displacement beyond the initial touch. Current profiles were classified using the following parameters: rapidly adapting (RA, inactivation time constant ($\tau$)>10ms), intermediately adapting (IA, 10ms < $\tau$<30ms), and slowly adapting (SA; $\tau$<30ms) current.

## Ex vivo teased nerve fiber recordings

Tibial skin nerve recordings were performed as previously described (*Reeh, 1988*; *Hoffman et al., 2018*). Briefly, animals were anesthetized and sacrificed via cervical dislocation. The leg of the animal

was shaved and the glabrous skin with the innervating tibial nerve was quickly removed and placed in a heated (32 +- 0.5 °C), oxygenated bath (pH 7.45 +- 0.05) consisting of (in mM): 123 NaCl, 3.5 KCl, 2.0 $CaCl_2$, 0.7 $MgSO_4$, 1.7 $NaH_2PO_4$, 5.5 glucose, 7.5 sucrose 9.5 sodium gluconate and 10 HEPES. Small nerve bundles were placed on a recording electrode and a blunt glass probe was used to search for receptive fields of single afferent fibers. Fibers were characterized based on their shape and conduction velocities: C-fibers<1.2 m/s; Aδ-fibers 1.2–10 m/s; and Aβ-fibers>10 m/s (**Koltzenburg et al., 1997**). Only slowly adapting Aβ and Aδ fibers were collected. Action potential thresholds were determined using a continuous force ramp (0–100 mN over 10 s). A custom designed feedback-controlled mechanical stimulator was used to stimulate the receptive fields with 2, 5, 10, 20, 40, 100, and 150 mN for 10 s. Sensitization was prevented by allowing 1 min breaks between mechanical stimulations. Data was recorded and analyzed with LabChart (ADInstruments; Colorado Springs, CO). A 3D printed plastic moat secured to the tissue with vacuum seal grease was used for recordings where Yoda1 was introduced to mechanically sensitive receptive fields. Drug was administered after determining the threshold with a continuous force ramp as described above, all other mechanical stimuli was conducted after addition of the drug. A 1 min recovery time was recorded to observe any drug induced activity.

## Data analysis

Histological comparisons were made using a two-way ANOVA. For calcium imaging data, the percentage of keratinocytes responding was compared via Chi square and post hoc Fisher's Exact tests. For behavior experiments, paw withdrawal thresholds and repeated stimulus responses were compared between two groups using non-parametric Mann-Whitney U-tests. Types of responses to the paintbrush and needle stimulus were analyzed using Chi square test with Fisher's exact tests. Spontaneous behavior was assessed using a Kruskal Wallis test or two-way ANOVA with Tukey's post-hoc test. Paw withdrawal latencies were compared between two groups using the Student's (two-tailed) *t* test. For high-speed imaging, percent responders to each stimulus were compared via Chi square and post hoc Fisher's Exact tests. Cumulative z-scores were analyzed using a two-way ANOVA with Bonferroni adjustment. Average z-score was compared using a Student's (two-tailed) *t*-test.

Skin nerve recordings were analyzed using a repeated measures two-way ANOVA with Sidak post-hoc test. Skin nerve mechanical thresholds were analyzed using Student's (two-tailed) *t* test. Patch clamp mechanical thresholds were analyzed using a Mann-Whitney U-test. Current amplitudes were analyzed using a non-parametric Mann-Whitney U-test. Percent responders to the patch clamp mechanical stimulus and current profile were compared using a Chi square and post hoc Fisher's exact tests.

For all behavior experiments, 'n' corresponds to the number of animals. For patch clamp studies, skin nerve recordings, or calcium imaging experiments at least n=3 animals were utilized for each group shown, and the n on the graph corresponds to the number of cells, fibers, or repetitions. For qPCR experiments, an n of three animals per group were utilized. Summarized data are reported as mean ± SEM. The number within the bars on the graph corresponds to the number of animals used. All data analyses were performed using Prism 7 software (GraphPad, La Jolla, CA), with an alpha value of 0.05 set a priori. *p<0.05, **p<0.01, ***p<0.001, ****p<0.0001, n.s. denotes a non-significant comparison.

## Acknowledgements

The authors thank Michael Lawlor, MD for assessing gross morphological differences between skin samples, as well as Reilly Allison and Sarah Langer for experimental assistance. The authors also thank the Medical College of Wisconsin Histology Core for tissue sectioning and staining, the Medical College of Wisconsin Imaging Core for slide scanning, and the Medical College of Wisconsin Tissue Bank for human skin tissues.

## Additional information

### Funding

| Funder | Grant reference number | Author |
|---|---|---|
| National Institute of Neurological Disorders and Stroke | NS040538 | Cheryl L Stucky |
| National Institute of Neurological Disorders and Stroke | NS070711 | Cheryl L Stucky |
| National Institute of Neurological Disorders and Stroke | NS108278 | Cheryl L Stucky |
| Medical College of Wisconsin | Advancing a Healthier Wisconsin Endowment | Cheryl L Stucky |
| National Institute of Neurological Disorders and Stroke | 1F31NS125941-01 | Alexander R Mikesell |

The funders had no role in study design, data collection and interpretation, or the decision to submit the work for publication.

### Author contributions

Alexander R Mikesell, Conceptualization, Data curation, Formal analysis, Validation, Investigation, Visualization, Methodology, Writing - original draft, Writing – review and editing; Olena Isaeva, Formal analysis, Validation, Investigation, Methodology, Writing – review and editing; Francie Moehring, Katelyn E Sadler, Conceptualization, Data curation, Formal analysis, Investigation, Visualization, Methodology, Writing – review and editing; Anthony D Menzel, Data curation, Formal analysis, Investigation, Methodology, Writing – review and editing; Cheryl L Stucky, Conceptualization, Supervision, Funding acquisition, Writing – review and editing

### Author ORCIDs

Alexander R Mikesell ⬤ http://orcid.org/0000-0002-8330-6608
Olena Isaeva ⬤ http://orcid.org/0000-0002-0587-6322
Francie Moehring ⬤ http://orcid.org/0000-0002-0071-5685
Katelyn E Sadler ⬤ http://orcid.org/0000-0003-2078-3527
Cheryl L Stucky ⬤ http://orcid.org/0000-0003-4966-6594

### Ethics

Animal experimentation: All protocols were in accordance with National Institutes of Health guidelines and were approved by the Institutional Animal Care and Use Committee at the Medical College of Wisconsin (Milwaukee, WI; protocol #383).

### Decision letter and Author response

Decision letter https://doi.org/10.7554/eLife.65987.sa1
Author response https://doi.org/10.7554/eLife.65987.sa2

## Additional files

### Supplementary files
• Transparent reporting form

### Data availability

All data generated or analyzed during this study are included in the manuscript and supporting files.

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
