## [Editor Report]

Although sensory neurons are thought to be the primary detectors of environmental stimuli in skin, it is more and more appreciated that non-neuronal cell types also play important roles. This study investigates whether a very common type of cell in the skin functions in touch sensation and identifies the mechanically gated ion channel Piezo1 as key gene.

---

## [Decision Letter]

**Decision letter after peer review:**

Thank you for submitting your article "Piezo1 mediates keratinocyte mechanotransduction" for consideration by *eLife*. Your article has been reviewed by 3 peer reviewers, including Alexander Theodore Chesler as Reviewing Editor and Reviewer #1, and the evaluation has been overseen by Kenton Swartz as the Senior Editor. The following individual involved in review of your submission has agreed to reveal their identity: Philippe Séguéla (Reviewer #3).

Overview of requested revisions (Recommendations for the authors):

Each reviewer had distinct suggestions for improving the manuscript including additional analyses of the datasets. Please refer to the individual reviews below and address as many of these comments as possible. If a particular comment cannot be addressed, please state the reason.

*Reviewer #1 (Recommendations for the authors):*

Summary:

Although sensory neurons are thought to be the primary detectors of environmental stimuli in skin, it is more and more appreciated that non-neuronal cell types also play important roles. Previous work from the Stucky group (and others) has shown stimulation of optical excitation of keratinocytes can evoke action potentials in sensory neurons and behavioural responses suggesting functional connectivity. Earlier work from the Stucky group provided evidence that keratinocytes are thermosenstive and required for normal temperature sensation. Here, they look into whether these cells are also important for mechanosensation. Moehring and colleagues convincingly show that keratinocytes have mechanically evoked currents mediated by Piezo1. They next provide evidence that removing Piezo1 from keratinocytes reduces the frequency of spiking in select types of sensory neurons to punctate and dynamic touch stimuli. Finally, they supply quite surprising data documenting significant behavioural deficits in Krt-conditional knockout mice.

Overall, an intriguing topic although I do have some questions. My biggest concerns is that the differences in the skin-nerve recordings are quite subtle whereas the behavioural effects are remarkably robust. How do the author reconcile this? Similarly, it is hard to understand how such profound mechanical deficits could be occurring given the known essential role for Piezo2 in many of these touch behaviours

Major Comments

1. This manuscript is lacking a discussion.

2. In the skin nerve prep experiments, the authors should examine the response of Abeta and ADelta fibres to Yoda1 application to the skin. If Piezo1-expressing keratinocytes strongly couple to these fibres, then Yoda1 application should cause AP firing selectively in large calibre mechanically-sensitive afferents. Given the relatively low expression of Piezo1 in these afferents, the Yoda1-induced firing should be abolished by keratinocyte-specific deletion of Piezo1. This experiment would strongly support the author's interpretation of their behavioural experiments using Yoda1.

3. Piezo2 is required in LTMRs for light touch sensation. Can the authors explain why Piezo1 in keratinocytes is not sufficient to support residual touch sensation in the afferent Piezo2 KO lines? If the authors interpretation is correct, the Yoda1 behavioural experiments imply that keratinocyte Piezo1 activation is sufficient to drive behavioural responses.

4. In the skin nerve prep experiments, SA-Abeta fibres showed deficient firing in the keratinocyte Piezo1 KO only at the very highest applied forces. Likewise for Adelta. The authors also show the SA-Abeta threshold was not altered. On the other hand, in the behavioural experiments the keratinocyte Piezo1 KO animals showed a remarkable impairment for both high-threshold AND light touch stimuli. How do the authors explain the reduced behavioural response to low-threshold stimuli given the unaltered afferent responses to stimuli of a similar type?

5. Figure 1C and 1E – the figures show that virtually 100% of mouse and human keratinocytes respond to the highest doses of Yoda1. The lack of response in the mouse KO to the highest Yoda1 dose indicates this must be Piezo1-mediated, and by implication 100% of keratinocytes express functional Piezo1. This seems very high and worth commenting on. Do the authors have immuno or RNAscope data to expect ubiquitous expression of Piezo1 in keratinocytes? The authors have previously shown that keratinocytes respond to cold temperatures. How can keratinocytes selectively regulate cold sensation when the same cold-sensing keratinocytes presumably also express Piezo1 and are mechanosensitive, and thus also couple to the mechanically sensitive afferents?

6. Similarly, keratinocytes respond to cold, hot and mechanical stimuli? How then would there be any selectivity at the sensory neuron level? Something doesn't quite make sense to me.

7. Figure 2A – the current clamp experiments need representative traces of the force-dependent membrane depolarization to gauge the shape/kinetics of the response. E.g. at the highest forces, do you see a VG calcium spike(let) that rides on top of the depolarization attributable to Piezo1 opening? (Would expect this given the calcium imaging!)

8. It is surprising that Piezo1 knockout diminishes the amplitude of mechano-currents but not the activation threshold- can the authors discuss what they think is happening here?

*Reviewer #2 (Recommendations for the authors):*

Figure 3 B-D. These results put together suggest that keratinocytes have MA currents with three different kinetics and PIEZO1 is responsible for the RA mediated 60-70% of all MA responses in keratinocytes, whereas the remaining ~30% of the MA currents are unaccounted for and are likely mediated by an unknown mechanosensor. This is an interesting and important observation. I think a little more information in this figure will be helpful. Specifically, the authors should consider showing representative traces for the IA and SA responses. Pannel C and D suggests that in addition to a decrease in RA currents, knocking out PIEZO1 results in smaller MA currents in the remaining IA/SA containing cells. Is this true? This a bit confusing and the authors should consider describing these panels better.

*Reviewer #3 (Recommendations for the authors):*

Moehring and collaborators report that the mechanosensitive channel Piezo1 is expressed in keratinocytes in mice and humans and claim that it contributes to touch sensation. The identification of Piezo1 as a major mechanotransducer in keratinocytes is convincing however proving its direct role in touch would require more solid experimental evidence.

Major issues

1) K14 expression is not exclusive to keratinocytes (refs 1,2), therefore both expression and conditional KO of Piezo1 in other K14-expressing epidermal cells (i.e. melanocytes, Langerhans cells) could contribute to the cellular and behavioral responses observed.

2) In absence of a functional link (release of a mediator?) demonstrated between activation of Piezo1 in keratinocytes and transduction in primary sensory neurons, the changes observed in the cKO mice could be due to developmental or homeostatic effects of the cKO in the epidermis and their indirect consequences on the mechanosensitivity of cutaneous sensory fibers. Supplemental figure 3 does not address potential changes in cellular composition of the epidermis in adult cKO mice.

3) Supplemental Figure 1. If Piezo1 is a major mechanotransducer in keratinocytes, it is not clear to me how its ablation does not have any measurable impact on their mechanical threshold.

4) Figure 3 A-F: the typical traces shown in A, C, E do not fit with quantitative data in B, D, F. This discrepancy is more obvious in the condition of 40 mN stimulations for the three types of fibers.

5) Figure 4J is missing.

6) The manuscript title does not reflect the complete story. The sections on mechanosensory fibers and touch are left out.

Other comments

1) Figure 1A shows a double normalization. The real expression level of Piezo1mRNA (i.e. relative to HPRT) in wildtype vs. cKO is not indicated.

2) Figure 1B. How many cells have been recorded from how many mice in both conditions?

3) Based on the currents (or absence of) shown in figures 2B and based on 2C, panel 2D does not fit, or this was plotted should be explained.

4) Figure 2C. How many cells have been recorded in both groups? Is there an effect of the cKO on the phenotypes of IA and SA currents?

5) Figure 3. Timescales missing in panels A-F.

6) Figure 4E. Does a stimulation with a needle ever evoke normal/innocuous, not always nocifensive responses?

7) The doses used for the agonist Yoda1 in vitro and in vivo are vastly different (Figures 1 and 4). Is it a typo?

8) Page 8. "Spontaneous behaviors" for behavioral outcomes elicited by Yoda1 injection is too vague.

References

1-Yoshimura N, Motohashi T, Aoki H, Tezuka K, Watanabe N, Wakaoka T, Era T, Kunisada T. Dual origin of melanocytes defined by Sox1 expression and their region-specific distribution in mammalian skin. Dev Growth Differ. 2013 Feb;55(2):270-81. doi: 10.1111/dgd.12034. Epub 2013 Jan 24. PMID: 23347447.

2-De La Cruz Diaz JS, Kaplan DH. Langerhans cells spy on keratinocytes. J Invest Dermatol. 2019 Nov;139(11):2260-2262. doi: 10.1016/j.jid.2019.06.120. PMID: 31648687; PMCID: PMC6818751.

[Editors’ note: further revisions were suggested prior to acceptance, as described below.]

Thank you for resubmitting your work entitled "Keratinocyte PIEZO1 mediates mechanosensation" for further consideration by *eLife*. Your revised article has been evaluated by Kenton Swartz (Senior Editor) and a Reviewing Editor.

The manuscript has been improved but there are just a few remaining issues that need to be addressed (see below). Most importantly, the reviewers all felt the work needed to be revised in light of the recent findings by Hill et al., that Piezo1 is expressed in sensory neurons and mediates mechanical itch.

*Reviewer #1 (Recommendations for the authors):*

The authors have done an admirable job addressing the reviewer's questions. Notably, the new data clarify important details.

We have a few remaining questions:

1. The revised patch clamp recordings cast doubt on the relevance of Piezo1 in keratinocyte mechanotransduction. Keratinocytes respond robustly to Piezo1 agonist (Figure 1C-F), yet they have very little Piezo1 transcript (Figure 1A) and are minimally affected by Piezo1cKO (Figure 2).

2. Is there a clearer effect of Piezo1cKO if the mechanical threshold and current amplitude (Figure 2B, F) are analyzed in separate groups of RA, IA, and SA currents?

3. Why do the representative RA currents run down with increasing membrane indentation (Figure 2A)?

4. The Yoda1-induced paw attending behavior (Figure 4B) is also hard to interpret – it seems unlikely to be due to the discussed mechanism of neuronal sensitization because the mechanical threshold for neuron firing is unchanged (Figure 4D)

5. Very recently it was shown that Piezo1 is functionally expressed in mouse NppB/Ssst positive neurons and drives mechanical itch responses. Could this be another explanation for the paw-attending behavior? Even though that work came out after the initial submission, it would be great if can be addressed and cited here.

6. Similarly, if Piezo1 is expressed in a subset of c-Fibers, is it surprising there seem to be no responses to Yoda in the skin-nerve recordings? Can this be discussed?

*Reviewer #3 (Recommendations for the authors):*

The authors have addressed reviewers' comments satisfactorily and the manuscript is significantly improved.

Due to recently published findings on the role of PIEZO1 in mechanical itch (Hill et al., 2022), they must now address the potential impact of the expression of PIEZO1 in subsets of primary somatosensory neurons on the interpretation of their data on Yoda1-evoked fiber firing activity and behavioral responses.

The title could be downtoned as keratinocyte PIEZO1 appears to be neither necessary nor sufficient for mechanosensation.

---

## [Author Response]

Reviewer #1 (Recommendations for the authors):1. This manuscript is lacking a discussion.

We have updated the manuscript to include a full Discussion section.

2. In the skin nerve prep experiments, the authors should examine the response of Abeta and ADelta fibres to Yoda1 application to the skin. If Piezo1-expressing keratinocytes strongly couple to these fibres, then Yoda1 application should cause AP firing selectively in large calibre mechanically-sensitive afferents. Given the relatively low expression of Piezo1 in these afferents, the Yoda1-induced firing should be abolished by keratinocyte-specific deletion of Piezo1. This experiment would strongly support the author's interpretation of their behavioural experiments using Yoda1.

To address this question, we performed teased fiber recordings while applying 1 mM Yoda1 to the receptive field of identified fibers. We found that Yoda1 application failed to induce any Yoda1-induced firing in any fiber types tested (Aβ n=10, Aδ n=10, C fibers n=10). In contrast, 1 μM capsaicin to the same receptive fields induced robust firing in all C fibers tested. These findings suggest that Yoda1 does not directly induce firing in primary afferent terminals, which is at odds with our original interpretation of the behavior data in Figure 4 where Yoda1 injection into the hind paw induced paw withdrawal behaviors. In light of this, we hypothesized that these behaviors induced by direct activation of PIEZO1 expressing cells were withdrawal behaviors elicited by Yoda1-mediated mechanical sensitization, such that the pressure of the glass floor on the paw skin was sufficient to induce withdrawal responses. To test this on a behavioral level, we measured von Frey withdrawal thresholds in mice following intraplantar injection of Yoda1. Yoda1 decreased the withdrawal thresholds of wildtype mice 30 min post injection but had no effect on PIEZO1cKO mice (Figure 4G). Next, we performed additional teased fiber recordings in the presence of 1 mM Yoda1 while simultaneously mechanically stimulating the receptive fields of identified fibers (Figure 4C-F). We found that Yoda1 elevated the mechanical firing frequency of C fiber types in wildtype preparations but had no effect on the mechanical responses of A fiber afferents. Interestingly, the Yoda1 induced mechanical sensitization of C fiber afferents was absent in PIEZO1cKO preparations, indicating a requirement for epidermal PIEZO1. These findings suggest that the Yoda1-induced behavioral responses were primarily due to the epidermal-PIEZO1 mediated sensitization of C fiber afferents.

3. Piezo2 is required in LTMRs for light touch sensation. Can the authors explain why Piezo1 in keratinocytes is not sufficient to support residual touch sensation in the afferent Piezo2 KO lines? If the authors interpretation is correct, the Yoda1 behavioural experiments imply that keratinocyte Piezo1 activation is sufficient to drive behavioural responses.

Based on our recently added teased fiber recordings (see comment 2 above), Yoda1 application is not sufficient to initiate firing in sensory fibers but does sensitize C fiber afferents to mechanical stimulation in an epidermal-PIEZO1 dependent manor. Therefore, we believe that both keratinocytes and sensory neuron mechanoreceptors function together to mediate touch sensation. Neither epidermal PIEZO1 deletion nor optogenetic inhibition is sufficient to completely abolish behavioral responses to mechanical stimulation, indicating the requirement of neuronal mechanotransducers in these behavioral responses. We hypothesize that activation of keratinocyte PIEZO1 amplifies (likely through the release of signaling mediators such as ATP) but does not necessarily initiate sensory fiber responses to mechanical force. In the absence of neuronal PIEZO2, the keratinocyte to neuronal mechanical signaling does not appear to be sufficient to on its own preserve behavioral responses to light touch.

4. In the skin nerve prep experiments, SA-Abeta fibres showed deficient firing in the keratinocyte Piezo1 KO only at the very highest applied forces. Likewise for Adelta. The authors also show the SA-Abeta threshold was not altered. On the other hand, in the behavioural experiments the keratinocyte Piezo1 KO animals showed a remarkable impairment for both high-threshold AND light touch stimuli. How do the authors explain the reduced behavioural response to low-threshold stimuli given the unaltered afferent responses to stimuli of a similar type?

We performed additional teased fiber recordings on all fiber types and our revised data in Figure 3 and figure 4 show that there is little effect of keratinocyte PIEZO1 deletion on afferent firing to mechanical stimulation of the receptive field. The only difference that remains is a decrease in PIEZO1cKO Aδ mechanically induced firing frequency, specifically at the highest forces tested. The dichotomy between the effect of PIEZO1 deletion on behavior and lack of effect on terminal firing is even more striking with the addition of these new data. A potential explanation is that the reflexive behavioral responses to touch may rely on the summation of activity in many overlapping receptive fields, and therefore, activity in keratinocytes from a broad area of skin. Alternatively, the mechanical responsiveness of a single afferent fiber may rely on far fewer keratinocytes. Another possibility is that the nature of our teased fiber recording preparation is not entirely representative of the processes that occur during in vivo mechanical stimulation; in our preparation the vasculature and connective tissues are removed and the corium (inside) of the epidermis is stimulated. In behavioral experiments, the stratum corneum (outermost layer) of keratinocytes was stimulated; perhaps there are anatomical differences in the ways in which keratinocytes respond to punctate stimuli in the two directional formats.

5. Figure 1C and 1E – the figures show that virtually 100% of mouse and human keratinocytes respond to the highest doses of Yoda1. The lack of response in the mouse KO to the highest Yoda1 dose indicates this must be Piezo1-mediated, and by implication 100% of keratinocytes express functional Piezo1. This seems very high and worth commenting on. Do the authors have immuno or RNAscope data to expect ubiquitous expression of Piezo1 in keratinocytes?

We performed RNAscope on skin sections and found that *Piezo1* mRNA is expressed throughout the epidermis in wild type skin and this staining is not present in the PIEZO1cKO skin. These findings are presented in Figure 1A. Although virtually 100% of keratinocytes responded during Yoda1 application, it is possible that some of these responses may not actually be due to Yoda1 but rather caused by the release of paracrine compounds from nearby cells that are activated by Yoda1.

6. Similarly, keratinocytes respond to cold, hot and mechanical stimuli? How then would there be any selectivity at the sensory neuron level? Something doesn't quite make sense to me.

Thank you for bringing up this interesting point. Based on previous work from our lab and others, we hypothesize that keratinocytes function as general detectors and amplifiers of cold, heat, and mechanical sensation by signaling to sensory neurons and thus increasing their likelihood of firing. For example, we have identified keratinocyte ATP release as important for normal behavioral responses to cold, heat and mechanical stimuli. Thus, epidermal purinergic signaling appears to be a ubiquitous mechanism that amplifies sensory neuron firing to multiple modalities of somatosensory stimuli. Detection of the sensory stimulus (cold, heat, mechanical) still requires the sensory fiber and it is likely at the level of the sensory neuron that sensory selectivity is mainly regulated. The keratinocytes seem to be generic amplifiers of somatosensory stimuli that function to potentiate but not necessarily initiate sensory neuron firing in response to heat, cold and mechanical stimuli. Additionally, there may exist modality-specific signaling pathways that convey specific aspects of cold, heat or mechanical stimuli to sensory neurons, although future studies are needed to identify these.

7. Figure 2A – the current clamp experiments need representative traces of the force-dependent membrane depolarization to gauge the shape/kinetics of the response. E.g. at the highest forces, do you see a VG calcium spike(let) that rides on top of the depolarization attributable to Piezo1 opening? (Would expect this given the calcium imaging!)

We had issues finding representative traces to clearly illustrate attributes of the mechanically induced current in keratinocytes, thus we decided to repeat these patch clamp experiments to confirm our original data set. Two skilled electrophysiologists in the lab have now independently observed results similar to those presented in updated Figure 2. Specifically, we show that PIEZO1cKO keratinocytes require a greater level of indentation to elicit mechanical currents compared to wild-type cells, indicating PIEZO1cKO keratinocytes have elevated mechanical thresholds (Figure 2B). In addition, there was an increase in the proportion of mechanically insensitive cells in PIEZO1cKO group compared to wild type (Figure 2C). In whole-cell current clamp mode we observed activation of fast conductance (presumably, calcium or sodium voltage-gated channels) in response to the indentation-induced membrane depolarization. We did not include these data in resubmission as we are still in the process of studying the effect of mechanical stimulation on keratinocytes membrane voltage and plan to describe these mechanisms in a future study.

8. It is surprising that Piezo1 knockout diminishes the amplitude of mechano-currents but not the activation threshold- can the authors discuss what they think is happening here?

We have updated the patch data in figure 2 and now find that PIEZO1 deletion increases the activation threshold of keratinocytes. However, the amplitude of the evoked mechanical current in keratinocytes did not show a clear dependence on the increase of membrane indentation depth. Figure 2A shows a representative example of a MA current evoked in a keratinocyte in response to stepwise increase in membrane indentation.

Reviewer #2 (Recommendations for the authors):Figure 3 B-D. These results put together suggest that keratinocytes have MA currents with three different kinetics and PIEZO1 is responsible for the RA mediated 60-70% of all MA responses in keratinocytes, whereas the remaining ~30% of the MA currents are unaccounted for and are likely mediated by an unknown mechanosensor. This is an interesting and important observation. I think a little more information in this figure will be helpful. Specifically, the authors should consider showing representative traces for the IA and SA responses. Pannel C and D suggests that in addition to a decrease in RA currents, knocking out PIEZO1 results in smaller MA currents in the remaining IA/SA containing cells. Is this true? This a bit confusing and the authors should consider describing these panels better.

Thank you for these comments. As mentioned in our responses to Reviewer 1, we re-evaluated the effects of PIEZO1 deletion on mechanically activated currents in keratinocytes. Example traces of different mechanical currents are now presented on Figure 2D. Our new data show that the percentage of cells that did *not* responded to membrane indentation is significantly increased in the PIEZO1cKO group compared to control. However, in this new data set, we did not observe any significant difference in the proportion of RA, IA, or SA currents evoked in PIEZO1cKO or wildtype keratinocytes.

Reviewer #3 (Recommendations for the authors):Moehring and collaborators report that the mechanosensitive channel Piezo1 is expressed in keratinocytes in mice and humans and claim that it contributes to touch sensation. The identification of Piezo1 as a major mechanotransducer in keratinocytes is convincing however proving its direct role in touch would require more solid experimental evidence.Major issues1) K14 expression is not exclusive to keratinocytes (refs 1,2), therefore both expression and conditional KO of Piezo1 in other K14-expressing epidermal cells (i.e. melanocytes, Langerhans cells) could contribute to the cellular and behavioral responses observed.

Although keratinocytes make up the vast majority (>95%) of epidermal cells, Reviewer 3 is correct in that other K14 expressing cells may be contributing the observed behavioral effects of the PIEZO1 knockout. We have updated the language in the manuscript when discussing the in vivo and ex vivo experiments to reflect this. We have also added to the discussion to better describe this limitation of the study.

2) In absence of a functional link (release of a mediator?) demonstrated between activation of Piezo1 in keratinocytes and transduction in primary sensory neurons, the changes observed in the cKO mice could be due to developmental or homeostatic effects of the cKO in the epidermis and their indirect consequences on the mechanosensitivity of cutaneous sensory fibers. Supplemental figure 3 does not address potential changes in cellular composition of the epidermis in adult cKO mice.

We agree and have updated our discussion to point out this limitation of our study. We tested the cold and heat sensitivities of wild type and PIEZO1cKO mice and found that there was no difference between the genotypes. Thus, any development or homeostatic effects of the knockout on behavior would have to be specific to mechanical sensitivity.

3) Supplemental Figure 1. If Piezo1 is a major mechanotransducer in keratinocytes, it is not clear to me how its ablation does not have any measurable impact on their mechanical threshold.

We conducted subsequent patch clamp recordings (see response to reviewer 1, comment 7 for more details) and now demonstrate a significant difference in mechanical threshold between the PIEZO1cKO keratinocytes and wildtype.

4) Figure 3 A-F: the typical traces shown in A, C, E do not fit with quantitative data in B, D, F. This discrepancy is more obvious in the condition of 40 mN stimulations for the three types of fibers.

We have updated the example traces to more accurately depict the quantitative data.

5) Figure 4J is missing.

Thank you for catching this. This is a typo, there is no figure 4J. We have updated to figure to correct this.

6) The manuscript title does not reflect the complete story. The sections on mechanosensory fibers and touch are left out.

We agree and have updated the title to Keratinocyte PIEZO1 mediates mechanosensation.

Other comments1) Figure 1A shows a double normalization. The real expression level of Piezo1mRNA (i.e. relative to HPRT) in wildtype vs. cKO is not indicated.

Thank you for pointing out this error. We have removed the double normalization, the figure now displays the expression level of PIEZO1mRNA relative to HPRT in wildtype vs. PIEZO1cKO keratinocytes.

2) Figure 1B. How many cells have been recorded from how many mice in both conditions?

For patch clamp experiments we used 5 mice per group. Data was collected from 37 cells in Piezo1 cKO group and 23 cells in wildtype group. This data is now presented in the figure 2 legend.

3) Based on the currents (or absence of) shown in figures 2B and based on 2C, panel 2D does not fit, or this was plotted should be explained.

Thank you for pointing this out. In our original submission, 2D included only the cells that responded to membrane indentation with mechanical currents (none of the mechanically insensitive cells). However, as explained in the response to reviewer 1 comment 7, we have had to redo this data and the new patch data is displayed in figure 2 of the manuscript.

4) Figure 2C. How many cells have been recorded in both groups? Is there an effect of the cKO on the phenotypes of IA and SA currents?

We recorded n=23 cells from wildtype tissue and n=37 cells from PIEZO1cKO tissue. 18 cells responded to mechanical stimulation in each group. The figure legend has been updated to include this information. We did not observe any effect of the PIEZO1 deletion on the phenotype of IA and SA currents.

5) Figure 3. Timescales missing in panels A-F.

Thank you for bringing this omission to our attention. The figure legend has been updated to include the timescale (10 seconds).

6) Figure 4E. Does a stimulation with a needle ever evoke normal/innocuous, not always nocifensive responses?

Yes, the needle stimulus sometimes evokes a simple withdrawal reflex (categorized as a normal response), a paw attending response (categorized as a nocifensive response), or the animal fails to respond to the needle stimulus (no response).

7) The doses used for the agonist Yoda1 in vitro and in vivo are vastly different (Figures 1 and 4). Is it a typo?

This is not a typo, we used higher doses for our in vivo experiments.

8) Page 8. "Spontaneous behaviors" for behavioral outcomes elicited by Yoda1 injection is too vague.

We agree and have updated the text to describe these as Yoda1 induced paw attending behaviors. Based on our Yoda1 skin nerve data (figure 4), we no longer consider these spontaneous behaviors, but rather the result of Yoda1 sensitizing the injected paw to mechanical stimulation, as described in our response to Reviewer 1 comment 1.

[Editors’ note: further revisions were suggested prior to acceptance, as described below.]

Reviewer #1 (Recommendations for the authors):The authors have done an admirable job addressing the reviewer's questions. Notably, the new data clarify important details.We have a few remaining questions:1. The revised patch clamp recordings cast doubt on the relevance of Piezo1 in keratinocyte mechanotransduction. Keratinocytes respond robustly to Piezo1 agonist (Figure 1C-F), yet they have very little Piezo1 transcript (Figure 1A) and are minimally affected by Piezo1cKO (Figure 2).

The reviewer is correct that we observed less robust PIEZO1 mRNA staining than would have been expected based on the calcium imaging data. This was surprising given that recent publications examining the role of keratinocyte PIEZO1 in wound healing (PMID: 34569935) and the epidermal mesenchymal transition (PMID: 35615675) both found more robust PIEZO1 expression in keratinocytes than we observed here. This discrepancy may reflect differences in the techniques used between these studies; we used RNAscope to measure PIEZO1 mRNA expression in the skin, whereas the other studies used PIEZO1 reporter lines and immunohistochemistry to visualize PIEZO1 expression in the epidermis. Thus, it is possible that the PIEZO1 mRNA staining we performed underrepresents the amount of PIEZO1 protein present in these cells. Ultimately, cellular function is a more sensitive reporter assay than mRNA transcript. Our data in the PIEZO1cKO keratinocytes shows no Yoda1-induced responses, indicating that the Yoda1-induced function is entirely due to PIEZO1.

For our patch clamp studies, the most noticeable effect of PIEZO1cKO was in the number of keratinocytes that *failed* to respond to the mechanical stimulus when it was applied (21.74% of the WT vs. 51.35% of the PIEZO1cKO cells were non-responsive to force). We also observed an increase in the mechanical threshold that elicited the first mechanical current upon membrane indentation in the PIEZO1cKO cells. However, we observed no change the maximum current amplitude evoked by membrane indentation in mechanically sensitive PIEZO1ckO keratinocytes compared to wildtype. This suggests that the role of PIEZO1 in keratinocyte mechanotransduction is in tuning the sensitivity of the cell to the mechanical stimulus but may be dispensable for the magnitude of inward current when the keratinocytes responds to force.

2. Is there a clearer effect of Piezo1cKO if the mechanical threshold and current amplitude (Figure 2B, F) are analyzed in separate groups of RA, IA, and SA currents?

All subtypes of mechanically induced current recorded in PIEZO1cKO keratinocytes showed a tendency for a higher threshold of current activation in response to the increase in membrane indentation compared to wildtype keratinocytes (please see data in (Author response table 1) ). The maximum amplitude was variable for all subtypes of current. We did not find any statistically significant difference between genotypes in these characteristics.

**Author response table 1. sa2table1:** 

	Type	WT	Piezo1cKO	P value
Threshold	RA	0.92±0.19 (n=9)	1.36±0.31 (n=9)	0.24
	IA	1.13±0.48 (n=4)	2.13±0.63 (n=2)	
	SA	0.85±0.22 (n=5)	1.86±0.48 (n=7)	0.13
MAX amplitude	RA	126.4±38.76 (n=9)	214±78.62 (n=9)	0.73
	IA	282.8±136.3 (n=4)	79.5±40.50 (n=2)	
	SA	320.8±164.8 (n=5)	128.1±48.03 (n=7)	0.22

3. Why do the representative RA currents run down with increasing membrane indentation (Figure 2A)?

We found that there was a large amount of variability in the current amplitude evoked by mechanical stimulation at each stimulus intensity in keratinocytes. As such, we did not observe a relationship between stimulus intensity and the amplitude of the evoked current. In some keratinocytes, the cell would respond with the largest inward current in response to the lowest membrane indentation and decrease in amplitude as the stimulus intensity was increased. In other keratinocytes, an intermediate intensity stimulus evoked the largest amplitude current. As such, the average maximum amplitude comparison in Figure 2C represents the maximum current amplitude evoked during the entire cell’s recording, regardless of stimulus intensity. These results in keratinocytes are very different to results from our recordings in isolated Dorsal Root Ganglion neurons where we observe a clear stimulus-intensity current-amplitude relationship using a similar stimulation protocol (PMID: 29563343). At this point, we do not have an explanation for the finding that keratinocytes do not respond in a stimulus-graded manner to mechanical poking. Because our data indicate that the mechanical sensitivity of keratinocytes is a result of the amalgamated activation of several channels, some of which we have not identified, we suggest that the channels involved may have different inactivation kinetics that could affect the keratinocytes’ total membrane response to increasing membrane indentation. Future studies are needed to determine whether this speculation is accurate or not.

4. The Yoda1-induced paw attending behavior (Figure 4B) is also hard to interpret – it seems unlikely to be due to the discussed mechanism of neuronal sensitization because the mechanical threshold for neuron firing is unchanged (Figure 4D)

We agree that the Yoda1-induced paw attending behavior is likely not due to direct neuronal sensitization by Yoda1 and have updated the text to better reflect and explain this. Since this attending behavior was reduced in the PIEZO1cKO mice, we hypothesize that Yoda1 is acting on epidermal PIEZO1 and sensitizing the keratinocytes to mechanical stimulation. This sensitization may lead to increased release of signaling molecules from keratinocytes, such as ATP, in response to mechanical stimulation, leading to the increased C fiber firing that we show in 4D. Since there was an increase in the firing frequency of wildtype C fibers treated with Yoda1 but not PIEZO1cKO C fibers, this may indicate that keratinocyte PIEZO1 mediated paracrine signaling is important for the depolarization of neighboring sensory afferents, leading to sustained sensory fiber firing in response to stimulation.

5. Very recently it was shown that Piezo1 is functionally expressed in mouse NppB/Ssst positive neurons and drives mechanical itch responses. Could this be another explanation for the paw-attending behavior? Even though that work came out after the initial submission, it would be great if can be addressed and cited here.

Thank you for suggesting this. We have updated the Discussion to include these recent findings. Although we cannot rule out the contribution of neuronal PIEZO1 to the Yoda1 paw attending behaviors, we believe our findings are specific to epidermal PIEZO1 since the PIEZO1 conditional keratinocyte knockout animals did not exhibit increased paw attending time following Yoda1 injection over vehicle treatment. This suggests that epidermal PIEZO1 is necessary for these responses, at least 10 minutes after Yoda1 injection.

6. Similarly, if Piezo1 is expressed in a subset of c-Fibers, is it surprising there seem to be no responses to Yoda in the skin-nerve recordings? Can this be discussed?

We agree that it is surprising that Yoda1 failed to induce firing in the skin nerve preparation, both in light of our calcium imaging data showing robust keratinocyte responses to Yoda1 (Figure 1C) and in light of the recent Hill et al., 2022 article demonstrating a role for neuronal PIEZO1 in mechanical itch. Because we applied Yoda1 to the dermal layer of the skin, it is possible that Yoda1 failed to penetrate sufficiently to the PIEZO1 expressing cells in the epidermis to induce sensory fiber firing. Interestingly, we observed an increase in C fiber mechanical sensitivity following Yoda1 application, an effect that was dependent on epidermal PIEZO1 expression. Therefore, an alternative explanation is that the Yoda1 induced behaviors may reflect increased hindpaw mechanical sensitivity, such that the innocuous force produced by the paw resting on the glass floor becomes sufficient to induce paw attending. In line with this hypothesis, we found that intraplantar injections of Yoda1 induced mechanical allodynia in wild-type but not PIEZO1cKO mice. Indeed, while Yoda1 can activate PIEZO1 channels on its own, it has been shown to prominently sensitize PIEZO1 to mechanical stimulation^26^. Such an increase in PIEZO1 mechanical sensitivity may explain why the effects of Yoda1 on sensory fiber firing were only observed in the presence of mechanical force.

Reviewer #3 (Recommendations for the authors):The authors have addressed reviewers' comments satisfactorily and the manuscript is significantly improved.Due to recently published findings on the role of PIEZO1 in mechanical itch (Hill et al., 2022), they must now address the potential impact of the expression of PIEZO1 in subsets of primary somatosensory neurons on the interpretation of their data on Yoda1-evoked fiber firing activity and behavioral responses.

We have updated our discussion of the Yoda1 evoked behavior and fiber sensitization to include this recent publication.

The title could be downtoned as keratinocyte PIEZO1 appears to be neither necessary nor sufficient for mechanosensation.

We agree and have updated the title to “Keratinocyte PIEZO1 modulates cutaneous mechanosensation”.